# The Function of MondoA and ChREBP Nutrient—Sensing Factors in Metabolic Disease

**DOI:** 10.3390/ijms24108811

**Published:** 2023-05-16

**Authors:** Byungyong Ahn

**Affiliations:** Department of Food Science and Nutrition, University of Ulsan, Ulsan 44610, Republic of Korea; byahn@ulsan.ac.kr

**Keywords:** MondoA, ChREBP, insulin resistance, obesity, diabetes, metabolic diseases

## Abstract

Obesity is a major global public health concern associated with an increased risk of many health problems, including type 2 diabetes, heart disease, stroke, and some types of cancer. Obesity is also a critical factor in the development of insulin resistance and type 2 diabetes. Insulin resistance is associated with metabolic inflexibility, which interferes with the body’s ability to switch from free fatty acids to carbohydrate substrates, as well as with the ectopic accumulation of triglycerides in non-adipose tissue, such as that of skeletal muscle, the liver, heart, and pancreas. Recent studies have demonstrated that MondoA (MLX-interacting protein or MLXIP) and the carbohydrate response element-binding protein (ChREBP, also known as MLXIPL and MondoB) play crucial roles in the regulation of nutrient metabolism and energy homeostasis in the body. This review summarizes recent advances in elucidating the function of MondoA and ChREBP in insulin resistance and related pathological conditions. This review provides an overview of the mechanisms by which MondoA and ChREBP transcription factors regulate glucose and lipid metabolism in metabolically active organs. Understanding the underlying mechanism of MondoA and ChREBP in insulin resistance and obesity can foster the development of new therapeutic strategies for treating metabolic diseases.

## 1. Introduction

Obesity has become a major public health concern over the past several decades. It is reported that more than 1.9 billion adults are overweight, with 650 million of these individuals being obese. Data also revealed that the global prevalence of obesity nearly tripled between 1975 and 2016 [1]. There are many factors that contribute to the rising prevalence of obesity, including changes in dietary habits, increased consumption of energy-dense and nutrient-poor foods, and decreased levels of physical activity. In addition, environmental and societal factors, such as changes in work and transportation patterns, may play a role in the development of obesity. Moreover, obesity is believed to be the primary driver of metabolic diseases, including insulin resistance, non-alcoholic fatty liver disease (NAFLD), atherosclerosis, and type 2 diabetes mellitus (T2DM) [2].

Insulin plays a crucial role in regulating glucose metabolism in various metabolic tissues, such as liver tissue, skeletal muscle, and adipose tissue. Insulin promotes glucose uptake and utilization in these tissues and suppresses hepatic glucose production. Insulin resistance occurs when these tissues become less responsive to insulin, leading to impaired glucose uptake and metabolism, and resulting in hyperglycemia [3]. Moreover, insulin resistance is a physiological state in which insulin-targeting tissues exhibit reduced responsiveness to high physiological levels of insulin. Insulin resistance often precedes the development of non-physiological elevated plasma glucose levels, which is the primary clinical symptom of T2DM. In the prediabetic state, insulin levels increase to maintain normal blood glucose levels. This chronic hyperinsulinemia can lead to beta cell exhaustion and failure, eventually resulting in T2DM [4,5]. In addition to T2DM, insulin resistance has been linked to other serious health conditions, including cardiovascular disease, hypertension, and certain types of cancer. The mechanisms underlying insulin resistance are complex and involve various factors, including genetics, lifestyle, and environmental exposures. While the precise causes of insulin resistance are not fully understood, it is believed to arise from a combination of factors, including inflammation, oxidative stress, and the accumulation of ectopic lipids in non-adipose tissues, such as those of the liver, heart, and skeletal muscle.

The prevention and management of insulin resistance and its associated diseases often involve lifestyle changes, including regular exercise, a healthy diet, and weight loss. Additionally, antidiabetic medications such as metformin, thiazolidinediones (TZDs), dipeptidyl peptidase 4 (DPP-4) inhibitors, and sodium glucose cotransporter 2 (SGLT2) inhibitors are currently used to improve insulin sensitivity and glycemic control in individuals with metabolic diseases [6]. However, further research is needed to discover new drug targets and to develop effective prevention and treatment strategies. Therefore, it is worthwhile to investigate MondoA and carbohydrate response element-binding protein (ChREBP) nutrient-sensing factors as potential therapeutic targets for protection against obesity, T2DM, and metabolic diseases.

## 2. Characteristics of MondoA and ChREBP

MondoA (also known as MLX-interacting protein, MLXIP) and ChREBP, also known as MLX-interacting protein-like (MLXIPL), are two paralogous transcription factors that play vital roles in glucose and lipid metabolism. MondoA and ChREBP are each a basic helix–loop–helix/leucine zipper (bHLH/LZ) transcription factor that forms a heterodimeric complex with Max-like protein X (MLX) to regulate gene expression in response to glucose and other nutrients [7,8,9]. MondoA regulates the expression of genes involved in glucose metabolism, glycogen synthesis, triglyceride synthesis, and insulin signaling, and it has been implicated in the development of metabolic diseases such as obesity, insulin resistance, and T2DM [10,11]. ChREBP, on the other hand, regulates the expression of genes involved in glucose and lipid metabolism, glycolysis, and de novo lipogenesis (DNL) in response to carbohydrate intake [12,13,14]. Although MondoA and ChREBP share structural and functional similarities, they have distinct physiological roles and target genes. The interplay between MondoA and ChREBP and their downstream targets is complex and not yet fully understood. Nevertheless, they are believed to play important roles in the regulation of glucose and lipid metabolism and the development of metabolic diseases. The general characteristics of MondoA and ChREBP are summarized in Table 1. In addition, results from mice models of knockout or overexpression of MondoA or ChREBP in different tissues are summarized in Table 2.

### 2.1. Regulation of MondoA Activity

MondoA was first discovered through yeast two-hybrid screening. It is predominantly expressed in skeletal muscle. In addition, it is ubiquitously expressed in other metabolically active tissues such as the heart, liver, adipose, kidneys, and pancreas [7] (Figure 1A). MondoA forms a complex with its binding partner, MLX to constitute the MondoA:MLX transcription factor complex [7]. This complex binds to a specific DNA sequence, called the carbohydrate response element (ChoRE), and activates the expression of target genes involved in glucose and lipid metabolism [10].

MondoA is a transcription factor that plays a vital role in glucose and lipid metabolism. MondoA functions as a glucose sensor and is activated by glucose metabolites, such as glucose-6-phosphate (G6P) and fructose-2,6-bisphosphate (Fru-2,6-BP) [15,16] (Figure 2A). While it is primarily responsive to glucose, evidence suggests that MondoA is also activated by certain non-glucose hexoses, such as allose and glucosamine [17]. Glucose metabolites are perceived by a glucose-sensing module (GSM) in the N-terminal region of MondoA, which consists of two domains: a low-glucose inhibitory domain (LID) and a glucose-response activation conserved element (GRACE). The GSM of MondoA depends on interactions with 14-3-3 proteins, which are a family of regulatory proteins that are involved in the regulation of a wide range of cellular processes, including metabolism and transcriptional regulation. Studies have shown that the interaction between MondoA and 14-3-3 proteins is dependent upon the phosphorylation of serine residues within the GSM [18,19].

MondoA has been shown to interact with the outer mitochondrial membrane in primary skeletal muscle cells. The interaction of MondoA with the outer mitochondrial membrane suggests that this interaction plays an important role in the regulation of mitochondrial metabolism and energy homeostasis. In addition, its association is involved in the coordination of the metabolic fluxes between the cytosol and mitochondria, which is important for the maintenance of energy homeostasis. The tricarboxylic acid (TCA) cycle is a key metabolic pathway that is involved in the production of ATP through oxidative phosphorylation in the mitochondria. The association of MondoA with the outer mitochondrial membrane has been implicated in the regulation of TCA cycle flux and mitochondrial respiration. Moreover, MondoA has been shown to directly regulate genes involved in glycolysis, which is the breakdown of glucose to produce energy, including lactate dehydrogenase A (LDHA), hexokinase II (HKII), and 6-phosphofructo-2-kinase/fructose-2,6-bisphosphatase 3 (PFKFB3) [20]. 

MondoA is a glucose-responsive transcription factor that is rapidly translocated to the nucleus in response to the metabolites of glucose intermediates such as G6P and Fru-2,6-BP. This translocation requires prior heterodimerization with its partner protein, MLX. Once MondoA–MLX heterodimers are formed, they bind to specific target gene promoters in a glucose-dependent manner, leading to the recruitment of histone acetyltransferases (HATs), such as p300/CREB-binding protein (CBP) [21]. These HATs modify the histone proteins that package DNA in the nucleus, resulting in a more open and accessible chromatin structure that allows for the binding of additional transcription factors and RNA polymerase II, ultimately leading to gene activation. The recruitment of the histone acetyltransferase by MondoA–MLX heterodimers facilitates the transcriptional activation of target genes that are involved in glucose and lipid metabolism, including those involved in glycolysis, lipogenesis, and glucose uptake. This coordinated regulation of gene expression by MondoA–MLX heterodimers is important for maintaining energy homeostasis and the proper functioning of metabolic pathways.

Glutamine is another metabolite that has been shown to modulate the activity of MondoA–MLX transcriptional complexes (Figure 2A). Specifically, in the presence of high levels of glutamine, MondoA–MLX heterodimers recruit a histone deacetylase (HDAC)-dependent co-repressor to target gene promoters, leading to transcriptional repression [17,22]. Under certain conditions, such as nutrient stress, or in cancer cells, glutamine can suppress the activity of MondoA and convert it to a transcriptional repressor. The mechanism by which glutamine regulates MondoA involves the recruitment of HDACs, which remove acetyl groups from histones and other proteins, leading to a more compact chromatin structure that is less accessible to transcription factors. The recruitment of HDACs by glutamine results in the formation of MondoA–MLX repressor complexes that bind to specific promoter sequences in target genes and suppress their expression.

MondoA is also sensitive to changes in cellular pH, with a low pH promoting its transcriptional activity (Figure 2A). A proposed mechanism for this effect is that the low pH stimulates ATP production by the mitochondrial respiratory chain, which in turn activates hexokinase, an enzyme that phosphorylates glucose to generate G6P in the cytoplasm [23]. Under conditions of cellular acidosis, such as during exercise or in cancer cells, the pH of the cytoplasm can decrease, leading to an increase in mitochondrial respiration and ATP production. This increase in ATP production can stimulate hexokinase activity, leading to the accumulation of G6P and promoting the transcriptional activity of MondoA.

MondoA has been shown to sense adenine nucleotides such as AMP, ADP, and ATP [24] (Figure 2A). Specifically, the binding of ATP to MondoA enhances its interaction with MLX and promotes the transcriptional induction of the target genes involved in glucose metabolism and energy homeostasis. The mechanism by which MondoA senses adenine nucleotides is not fully understood. Nonetheless, it has been suggested that changes in the intracellular levels of these molecules may affect the redox state of the cell and/or the activity of signaling pathways that regulate MondoA activity. For example, it has been suggested that the binding of ATP to MondoA may promote the production of reactive oxygen species (ROS), which in turn activates the downstream signaling pathways that promote the transcriptional induction of target genes. In addition to adenine nucleotides, MondoA has also been reported to sense other adenosine-containing molecules such as S-adenosylmethionine (SAM), a molecule involved in the transfer of methyl groups in cellular processes [25]. MondoA has been shown to regulate the metabolism of SAM in the liver, suggesting a role in the regulation of methylation reactions.

Mammalian target of rapamycin (mTOR) is a key nutrient sensor and signaling molecule that regulates various cellular processes, including cell growth, lipid metabolism, and autophagy. Recent studies have demonstrated that mTOR can interact with MondoA and modulate its transcriptional activity. mTOR can directly phosphorylate MondoA and promote its degradation, which can lead to a decrease in MondoA-mediated transcriptional activation. In addition, mTOR can interact with MondoA in a more direct manner through physical interaction between the two proteins. This interaction can have a suppressive effect on MondoA transcriptional activity, potentially by interfering with the binding of MondoA–MLX complexes to target gene promoters [26].

MondoA is a key nutrient sensor that responds to changes in levels of glucose, glucose metabolites, adenine nucleotides, and adenosine-containing molecules. To the best of our knowledge, there are no reports suggesting that fatty acids or amino acids directly regulate MondoA levels or activity. However, it is important to note that fatty acids and amino acids can indirectly affect MondoA activity through their effects on upstream signaling pathways and metabolites. For example, fatty acids and amino acids can activate mTOR, which can interact with MondoA and modulate its transcriptional activity, as discussed above. Additionally, amino acids can control the activity of other transcription factors, which can interact with MondoA and modulate its transcriptional activity.

### 2.2. Function of MondoA in Skeletal Muscle

Skeletal muscle plays a crucial role in the regulation of whole-body glucose homeostasis. Skeletal muscle is the largest organ in the body, comprising approximately 40% of total body mass. It is also a major site of glucose uptake and utilization in the body, accounting for up to 80% of insulin-stimulated glucose uptake in healthy individuals. Hence, the efficient regulation of glucose uptake and utilization in skeletal muscle is critical for maintaining whole-body glucose homeostasis and preventing the development of metabolic disorders such as insulin resistance and T2DM [27,28]. Regular exercise and enhanced skeletal muscle metabolism have a positive impact on metabolic health and can reduce the incidence of various metabolic diseases, including insulin resistance, cardiovascular disease, and hepatic steatosis. Moreover, regular exercise can improve glucose uptake and utilization in skeletal muscle, leading to improved insulin sensitivity and glucose homeostasis. Exercise can also promote the breakdown of stored fat in adipose tissue and the liver, reducing the risk of hepatic steatosis and other metabolic diseases [29].

During physical activity, skeletal muscle has an increased energy demand, and glucose uptake is rapidly upregulated to provide the necessary fuel for muscle contraction. This process is tightly regulated by a complex network of signaling pathways and transcriptional regulators, including MondoA and its binding partner, MLX. However, the specific roles of MondoA in skeletal muscle are not yet fully understood. Interestingly, MondoA-knockout mice have enhanced glycolytic rates, which is likely owing to an increase in glucose uptake in skeletal muscle resulting from the loss of MondoA-mediated glucose sensing and signaling pathways [30]. In the presence of high levels of glucose and fructose, MondoA activates the transcription of the thioredoxin-interacting protein (TXNIP) and arrestin domain-containing 4 (ARRDC4) [11,17,31]. Both TXNIP and ARRDC4 are potent inhibitors of insulin signaling and glucose uptake by reducing the cell surface expression of glucose transporters and inhibiting the activity of glucose transporters [32,33,34]. Moreover, MondoA plays an important role in regulating the glycogen synthesis pathway by promoting the gene transcription of phosphoprotein phosphatase 1 regulatory subunit 3A (PPP1R3A) and phosphoprotein phosphatase 1 regulatory subunit 3B (PPP1R3B), which are regulatory subunits of glycogen synthase [11,35]. Muscle-specific MondoA ablation in mice decreases the glycogen level in the skeletal muscle [34] (Table 2).

In addition to insulin signaling and glucose and glycogen metabolism, MondoA can enhance the genes involved in lipid metabolism, including fatty acid thioesterification (ACSL1 and ACSL4), desaturation (FADS1, FADS2, SCD1, and SCD5), elongation (ELOVL5 and ELOVL6), and TAG synthesis (DGAT1 and DGAT2). MondoA can also promote the hexosamine biosynthetic pathway, such as glutamine-fructose-6-phophate transaminase (GFPT1 and GFPT2) [11]. MondoA knockdown in human skeletal myotubes reduces oleate-induced TAG accumulation and activates glucose uptake [10]. Muscle-specific MondoA deficiency decreases muscle TAG accumulation in high-fat-diet (HFD)-fed mice and promotes insulin signaling and glucose uptake in skeletal muscle [11] (Table 2). Therefore, MondoA, a nutrient-regulated transcription factor, acts as a metabolic gatekeeper in muscle and plays a crucial role in regulating glucose uptake and utilization in muscle. However, further research is needed to fully understand the mechanisms underlying these effects and the context-dependent roles of MondoA in different tissues and under different metabolic conditions.

### 2.3. Regulation of ChREBP Activity

ChREBP was first identified in rat liver as a transcription factor that binds to and activates the expression of the liver-type pyruvate kinase gene (Pklr) [8]. Pklr is a key enzyme in the glycolytic pathway that converts glucose to pyruvate. Activation of Pklr expression by ChREBP enables increased glucose utilization and storage in the liver. ChREBP is primarily expressed in the liver, but it is also found in other tissues, such as adipose tissue, the kidneys, pancreas, and small intestine [36,37,38,39] (Figure 1B). The expression of ChREBP is relatively low in skeletal muscle and the heart [39].

Like MondoA, ChREBP is a transcription factor that plays a critical role in the regulation of carbohydrate metabolism and lipid synthesis. ChREBP is primarily regulated by glucose metabolites, such as glucose-6-phosphate (G6P), xylulose-5-phosphate (Xu5P), and fructose-2,6-bisphosphate (F2,6-BP) [40,41,42,43] (Figure 2B). When glucose levels are high, ChREBP is activated and translocates to the nucleus. Then, ChREBP forms a heterodimerization with MLX, and this complex (ChREBP:MLX) binds to a specific DNA sequence called carbohydrate response elements (ChoREs) in target genes [9]. The nuclear import of ChREBP is mediated by importin-α, a protein that recognizes and binds to nuclear localization signals (NLS) on cargo proteins and facilitates their transport into the nucleus [44,45]. The nuclear export and cytoplasmic retention of ChREBP are regulated by two key proteins: chromosome region maintenance protein 1 (CRM1) and 14-3-3 proteins [18,46,47,48]. The nuclear export of ChREBP is mediated by CRM1, which recognizes the nuclear export signal (NES) located within the bHLH domain of ChREBP. The binding of CRM1 to ChREBP promotes its transport out of the nucleus and into the cytoplasm. ChREBP contains multiple phosphorylation sites that can be recognized by 14-3-3 proteins. The binding of 14-3-3 to ChREBP promotes its retention in the cytoplasm, preventing its entry into the nucleus and transcriptional activity. Upon activation by glucose or nutrient signals, ChREBP regulates the expression of genes involved in glycolysis, DNL, and glucose uptake in the liver and other tissues. When glucose availability is limited, ChREBP activity can be inhibited by branched-chain keto-amino acids (BCKAs) and fatty acids, thereby reducing glucose utilization and promoting alternative energy sources such as ketone bodies [49,50] and polyunsaturated fatty acids [51] (Figure 2B). The regulation of ChREBP activity by nutrient molecules is a complex process that involves multiple signaling pathways and feedback mechanisms [40,41,42,43]. The balance between ChREBP activation and inhibition is critical for maintaining energy homeostasis and preventing metabolic diseases.

### 2.4. Function of ChREBP in Liver

The liver is a major organ involved in glucose homeostasis. It plays a crucial role in regulating systemic glucose levels in response to different energy states [52]. In the fed state, when glucose and other nutrients are abundant, insulin promotes glucose uptake and storage in peripheral tissues such as muscle and adipose tissues [53]. In the liver, insulin also suppresses glucose production by inhibiting gluconeogenesis and promoting glycogen synthesis. In contrast, during fasting or energy deprivation, insulin levels decrease and counter-regulatory hormones such as glucagon and cortisol increase, promoting the release of glucose from liver glycogen stores and stimulating gluconeogenesis to maintain blood glucose levels [54]. The liver also breaks down fatty acids to release energy for other tissues through β-oxidation and produces ketone bodies that can be used as an alternative energy source by other organs.

ChREBP exists in two isoforms, alpha and beta, which are transcribed from alternative exon 1a or 1b to exon 2 [36]. ChREBPα is the primary isoform and is predominantly expressed in the liver. ChREBPβ is expressed at lower levels than ChREBPα and is found in a wider range of tissues, including adipose tissue, muscle, and the pancreas. Both ChREBP α and β are activated by glucose and other sugars, which stimulate their translocation to the nucleus and binding to DNA at specific ChoREs in the promoter regions of target genes. Although ChREBPβ has a similar function to ChREBPα in the regulation of glucose and lipid metabolism, its precise role in these processes is not yet fully understood. Some studies suggest that ChREBPβ may play a more important role in the regulation of insulin sensitivity than ChREBPα, and that dysregulation of ChREBPβ may contribute to the development of insulin resistance and T2DM [55,56,57].

ChREBP has been shown to regulate the expression of several key enzymes involved in glycolysis and lipogenesis in the liver. These include enzymes involved in glycolysis, such as liver-type pyruvate kinase (L-PK), and enzymes involved in lipogenesis, such as acetyl-CoA carboxylase (ACC), fatty acid synthase (FAS), ATP-citrate lyase (ACLY), stearoyl-CoA desaturase-1 (SCD1), and glycerol-3-phosphate dehydrogenase (GPDH) [12,58,59,60]. Global-ChREBP-deficient mice exhibit reduced expression of key lipogenic genes such as ACC, FAS, ACLY, and SCD1, resulting in decreased lipid synthesis and accumulation in the liver under various nutrient conditions [61,62,63,64,65] (Table 2). Liver-specific ChREBP-deficient mice also show improved hepatic and peripheral insulin sensitivity [37,66,67] (Table 2). Conversely, in mice fed a high-fat diet, adenoviral-mediated overexpression of ChREBP in the liver improves glucose tolerance and insulin sensitivity, although hepatic steatosis is induced through activation of FAS, ACC, and SCD1 expression. ChREBP expression is upregulated in the liver and is positively correlated with the degree of hepatic steatosis in patients with non-alcoholic steatohepatitis. However, ChREBP expression is inversely correlated with insulin resistance [68]. Interestingly, some recent studies showed that ChREBP directly regulates the gene expression of fibroblast growth factor 21 (FGF21) [69,70,71]. FGF21, an important hormone, has therapeutic potential for the treatment of obesity, T2DM, and other metabolic diseases [72,73].

### 2.5. Function of ChREBP in Adipose Tissue

Adipose tissue (AT) is a specialized connective tissue that stores excess energy in the form of triglycerides and releases fatty acids and glycerol during energy expenditure. Adipose tissue is not only a passive energy storage site but also an active endocrine organ that secretes a variety of hormones and cytokines, collectively known as adipokines, which play important roles in regulating whole-body energy and glucose homeostasis [74,75,76]. Adipose tissue has two main types of adipose tissue, white adipose tissue (WAT) and brown adipose tissue (BAT). WAT is the primary site of energy storage in the body, and it functions to store excess energy in the form of triglycerides during periods of caloric excess. In contrast, during periods of energy deficiency, WAT releases fatty acids and other metabolites into the circulation for use as energy sources by other tissues [77,78]. BAT, on the other hand, is specialized for the generation of heat through a process known as non-shivering thermogenesis. BAT contains a large number of mitochondria and expresses high levels of uncoupling protein 1 (UCP1), which uncouples mitochondrial respiration from ATP synthesis and generates heat instead. BAT is important for the maintenance of body temperature in response to cold exposure and can also contribute to energy expenditure and metabolic homeostasis [79,80,81].

In adipose tissues, ChREBP controls the expression of genes involved in lipogenesis. Adipocyte-specific knockout of ChREBP induces whole-body insulin resistance, impaired GLUT4 (glucose transport 4) translocation and exocytosis, and inflammation [82] (Table 2). In contrast, transgenic overexpression of constitutively active ChREBP (ChREBP-CA) in adipose tissues increases the expression of key enzymes involved in lipogenesis; however, it remarkably protects against diet-induced obesity and insulin resistance [83]. Notably, ChREBP expression in adipose tissues is profoundly decreased in Glut4-lacking mice. Adipose-tissue specific overexpression of GLUT4 regulates the expression of ChREBP and lipogenic target genes. Glucose-mediated activation of ChREBPα also induces the expression of ChREBPβ [36]. ACLY-deficient adipocytes have shown that both ChREBPα and β expression are significantly reduced. In addition, in adipocyte-specific ACLY-deficient mice, insulin resistance is developed in females but not in males [84]. Adipose-tissue Rictor (mTORC2 subunit)-deficient mice show reduced expression of ChREBPβ and genes involved in lipogenesis [85]. In addition to regulating glucose metabolism and lipogenesis, ChREBP has been shown to regulate the expression of genes involved in adipocyte differentiation [86] and white adipocyte browning [87]. ChREBP deficiency appears to impair the ability of BAT to respond to excess energy intake and maintain body temperature [61]. Moreover, glucose and triiodothyronine, which are known to stimulate BAT activity, together elevate the ChREBP-mediated gene expression of Ucp1 in brown adipocytes [87]. ChREBP induces PPARγ activity, and its target genes are involved in thermogenesis, including Ucp1 and Cidea, in adipocytes and WAT. Conversely, overexpression of ChREBPβ, known as a constitutively active form of ChREBP, inhibits the gene expression of thermogenic factors such as Ucp1 and Dio2 and leads to BAT whitening [88]. Therefore, further studies are needed to fully elucidate the role of ChREBP in WAT and BAT.

### 2.6. Function of ChREBP in Pancreas

Pancreatic beta-cells, which are specialized cells located in the pancreas, are responsible for producing and secreting insulin and play a critical role in the regulation of glucose homeostasis. Insulin secretion from beta-cells is tightly regulated and is responsive to changes in nutrient availability, such as an increase in circulating glucose levels after a meal. When glucose levels rise, beta-cells respond by secreting insulin, which stimulates glucose uptake and utilization by tissues such as muscle and adipose tissue [89]. When beta-cell function is impaired, such as in the case of T2DM, insulin secretion is decreased or impaired, which can lead to hyperglycemia (high blood glucose levels) [90]. Therefore, the proper functioning of pancreatic beta-cells is essential for the maintenance of glucose homeostasis and the prevention of insulin resistance, diabetes, and metabolic diseases.

ChREBPα and ChREBPβ are not only expressed in pancreatic β-cells but are also necessary for glucose-stimulated β-cell proliferation [91,92]. Pancreatic beta-cell proliferation and regeneration are important processes that contribute to the maintenance of normal beta-cell mass and function, which are essential for the regulation of glucose homeostasis. Overexpression of ChREBPα elevates the expression of cell-cycle regulators, including cyclin D2, cyclin A, cyclin E, and cyclin-dependent kinase (CDK) 4/6 [91]. Knockdown of ChREBPβ inhibits the expression of cyclin A and cyclin E, leading to blocking glucose-induced β-cell proliferation [92]. Two recent studies revealed that retinoic acid receptor-related orphan receptor-γ (RORγ, known as a driver of the cell cycle) and nuclear factor erythroid 2-related factor 2 (NRF2, known as a critical transcription factor for antioxidant enzymes) are required for glucose-induced and ChREBP-mediated β-cell proliferation [93,94]. Therefore, these results demonstrate that ChREBP is a vital regulator and a potential therapeutic target for β-cell regeneration. 

Interestingly, ChREBPβ expression is increased in diabetic mice models [95]. Overexpression of ChREBP causes β-cell apoptosis in vitro and in vivo, promotes the accumulation of lipids in β-cells, and reduces mitochondrial fatty acid β-oxidation [63]. Under conditions of chronic caloric excess, ChREBP promotes the accumulation of lipids in pancreatic β-cells, which contributes to the development of β-cell dysfunction (Table 2). Moreover, activation of ChREBP induces the expression of TXNIP [63,96], a crucial molecule involved in inflammation, oxidative stress, and apoptosis in β-cells [97,98]. The ablation of TXNIP protects against glucotoxicity-induced β-cell apoptosis [97]. Accordingly, ChREBP represents a potential target for interventions aimed at preventing or treating β-cell dysfunction in the context of chronic caloric excess and obesity-related metabolic diseases.

### 2.7. Function of TXNIP as a Target of MondoA and ChREBP

TXNIP is a protein that regulates the activity of the antioxidant thioredoxin (TRX) by binding to it and inhibiting its activity [99]. TXNIP is involved in various cellular processes, including redox signaling, glucose metabolism, and apoptosis [100]. TXNIP has been implicated in the development of several diseases, including cancer, diabetes, and cardiovascular disease [101,102]. In diabetes, TXNIP has been shown to play a role in the regulation of glucose homeostasis by modulating the activity of pancreatic beta cells, which produce insulin [100]. Under the condition of high glucose, TXNIP expression is increased, leading to the inhibition of insulin secretion and promotion of beta cell apoptosis. This can contribute to the development of hyperglycemia and T2DM. TXNIP, on the other hand, is a protein that is involved in the regulation of glucose uptake and insulin signaling [31]. It acts as a negative regulator of glucose uptake by binding to Glut1 and Glut4 transporters, which are responsible for glucose transport into cells [33,103]. TXNIP binds to fructose transporters, Glut2 and Glut5, leading to promote fructose absorption [32]. TXNIP also inhibits insulin signaling by binding to and inhibiting the activity of Akt, a key downstream effector of insulin signaling [104]. TXNIP has also been linked to the development of cardiovascular disease, as it can promote oxidative stress and inflammation in the vascular endothelium, leading to endothelial dysfunction and atherosclerosis [105,106]. Furthermore, TXNIP has been shown to play a role in the regulation of cell proliferation and apoptosis in cancer cells. The precise mechanisms underlying the role of TXNIP in these diseases are still being investigated, and its potential as a therapeutic target is currently being explored.

### 2.8. Post-Translational Modification of MondoA and ChREBP

Post-translational modification (PTM) refers to the covalent modification of proteins after their synthesis. PTMs alter the chemical, physical, and functional properties of proteins and play a crucial role in the regulation of protein function, stability, localization, and interaction with other proteins and cellular components. There are many types of PTMs, including phosphorylation, glycosylation, acetylation, methylation, ubiquitination, O-GlcNAcylation, and many others. Each of these modifications involves the covalent addition or removal of specific chemical groups from amino acid residues in the protein [107,108,109,110]. PTMs of ChREBP have been demonstrated to play a critical role in regulating its activity in different cellular and nutritional contexts. Phosphorylation of ChREBP by protein kinase A (PKA) or AMP-activated protein kinase (AMPK) inhibits the activity of ChREBP by promoting its cytoplasmic localization and preventing its nuclear translocation [111,112] (Figure 2). Acetylation of ChREBP by p300 [113,114], or O-GlcNAcylation of ChREBP by O-GlcNAc transferase (OGT) [115], or host cell factor 1 (HCF-1) [116] enhances ChREBP transcriptional activity and promotes its nuclear localization (Figure 2). However, there have been no studies on PTMs of MondoA. Therefore, given the importance of MondoA in the regulation of glucose and lipid metabolism, further research is needed to elucidate the regulatory mechanisms of MondoA in response to nutrient levels.

**Table 1 ijms-24-08811-t001:** General characteristics of MondoA and ChREBP.

Feature	MondoA	ChREBP	References
Gene symbol and other names	*MLXIP* (MLX-interacting protein)	*MLXIPL* (MLX-interacting protein-like) MondoB	[7,8,9]
Isoforms and protein size	MondoA (919 AA)	ChREBP-α (852AA) ChREBP-β (675AA)	[7,36,117]
Mainly enriched organ	Skeletal muscle	Liver Adipose tissue	[7,36,37,38,39]
Major pathway	Insulin signaling Glucose transporter	De novo lipogenesis Glycolysis	[10,11,12,13,14]
Post-translational modification	Not available	Phosphorylation Acetylation *O*-GLcNAcylation	[111,112,113,114,115,116]
Inhibitor	SBI-477/993	Not available	[10]

**Table 2 ijms-24-08811-t002:** Summary of MondoA and ChREBP roles in experimental mice models.

Experimental Models	Context	Diet Duration	Body Weight	Fat Mass	Hepatic Steatosis	Insulin Resistance	References
Whole-body MondoA knockout	Chow diet		NC	-	-	NC	[30]
Muscle-specific MondoA knockout	Chow diet		NC	-	NC	NC	[11]
	60% HFD	16 weeks	NC	-	NC	Improved	[11]
Whole-body ChREBP knockout	Chow diet		NC	NC or down	NC or improved	Moderately induced	[61,62,64]
	High-starch diet	1 week	NC	NC	Improved	Moderately induced	[61]
	High-fructose diet (70%)	3 weeks	Down	-	Improved	NC	[65]
	Western diet (41% Fat)	14 weeks	Down	Down	Improved	-	[64]
Whole-body ChREBP knockout in ob/ob mice	Chow diet		Down	Down	Improved	Moderately improved	[62]
Liver-specific ChREBP knockout	Chow diet		NC	Down	NC	Moderately improved	[67]
	45% HFD	12 weeks	NC	NC	NC	Improved	[67]
	High-carbohydrate diet (70%)	12 weeks	Down	Down	Improved	Moderately improved	[67]
	High-fructose diet (60%)	9 weeks	Down	Down	NC	Improved	[37]
Liver-specific ChREBP knockdown in ob/ob mice	Chow diet		Down	Down	Improved	Improved	[62]
Liver-specific ChREBP expression	Chow diet		NC	Down	Induced	NC	[68]
	60% HFD	10 weeks	NC	Down	Induced	Improved	[68]
Adipose-tissue-specific ChREBP knockout	Chow diet		NC	NC	Improved	Induced	[82]
	55% HFD	16 weeks	NC	NC	NC	Induced	[82]
Adipose-tissue-specific ChREBP expression	Chow diet		NC	NC	NC	NC	[83]
	60% HFD	10 weeks	Down	Down	Improved	Improved	[83]
Pancreatic b-cell-specific ChREBP expression	Chow diet		Down	-	-	Induced	[63]
SBI-993 (MondoA inhibitor) administration	60% HFD	8 weeks	Down	-	Improved	Improved	[10]

“-” means not measured, NC means not changed.

### 2.9. Development of Therapeutics for MondoA and ChREBP

The development of therapeutics targeting MondoA and ChREBP is an active area of research, given their important roles in regulating glucose and lipid metabolism and their potential as targets for the treatment of diabetes, obesity, and metabolic diseases. One potential strategy for targeting MondoA and ChREBP is to develop small-molecule inhibitors or activators that can control their activity. Another approach is to target the upstream regulators of MondoA and ChREBP, such as AMP-activated protein kinase (AMPK) and protein kinase A (PKA), which are known to regulate their activity. There is also interest in developing gene therapy approaches to modulate the expression of MondoA and ChREBP in specific tissues. This could involve the use of viral vectors to deliver therapeutic genes to target tissues, or the use of CRISPR/Cas9 gene editing technology to selectively modulate the expression of MondoA and ChREBP in specific tissues. At present, there is only one compound, called SBI-477/993, that has been developed to inhibit MondoA activity [10]. SBI-477/993 is a small-molecule inhibitor that was identified through a high-throughput screening approach. SBI-477/993 has been shown to promote glucose uptake in skeletal muscle and improve glucose tolerance and insulin resistance in mice fed an HFD. SBI-477/993 also can regulate ChREBP activity [10]. However, it is important to note that SBI-477/993 is still in the early stages of development, and further studies will be needed to determine its efficacy and safety.

## 3. Conclusions and Future Direction

The nutrient-sensing factors MondoA and ChREBP play important roles in the regulation of metabolic glucose homeostasis and insulin sensitivity. They also have distinct roles in the regulation of glucose and lipid metabolism in different tissues. MondoA is predominantly expressed in skeletal muscle and is regulated by glucose and other carbohydrates. MondoA has been shown to repress the expression of genes involved in glucose metabolism, such as the glycogen-associated form of protein phosphatase 1 (PP1G), Krüppel-like factor 10 (KLF10) and KLF11, and to inhibit genes involved in triglyceride (TAG) synthesis and remodeling. Recent studies revealed that MondoA directly regulates several genes involved in the metabolic pathways of fatty acid thioesterification (Acsl3, Acsl4), elongation (Elovl5, Elovl6), desaturation (Scd1, Fads1), and TAG synthesis (Dgat1, Dgat2) in skeletal muscle. Therefore, MondoA acts as a negative regulator of fuel metabolism in skeletal muscle. ChREBP, on the other hand, is primarily expressed in the liver and adipose tissue, and promotes de novo lipogenesis in these tissues. ChREBP is also activated by glucose and other carbohydrates and induces the expression of genes involved in de novo lipogenesis and triglyceride synthesis. In addition, ChREBP regulates glucose metabolism by promoting the expression of genes involved in glycolysis and glucose uptake in adipose tissue. Although there is a growing body of evidence suggesting that MondoA and ChREBP transcription factors play important roles in the regulation of glucose and lipid metabolism, much remains to be understood about their functions and effects in different contexts. 

Future studies should focus on using more specific and rigorous models to investigate the functions of MondoA and ChREBP in different tissues and under different physiological and pathological conditions. In addition to tissue-specific knockout or overexpression models, advanced technologies such as CRISPR/Cas9-mediated gene editing, single-cell RNA sequencing, and metabolic flux analysis could be used to provide a more detailed understanding of the regulatory networks in which MondoA and ChREBP participate. Furthermore, it is important to investigate the potential interactions and crosstalk between MondoA, ChREBP, and other key regulators of metabolism, such as metabolic pathways and other transcription factors. Accordingly, these pathways and regulators could provide important insights into the mechanisms underlying the regulation of metabolism and could support the development of new therapeutic strategies to improve treatment of obesity, insulin resistance, T2DM, and metabolic diseases.

## Figures and Tables

**Figure 1 ijms-24-08811-f001:**
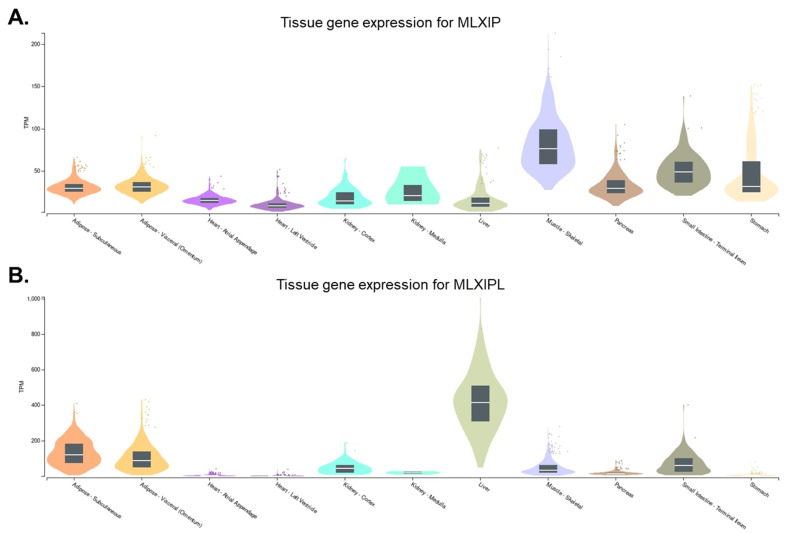
Gene expression levels of MLXIP and MLXIPL in different tissues. The tissue expression pattern of (**A**) MLXIP (https://gtexportal.org/home/gene/MLXIP (accessed on 12 May 2023)) and (**B**) MLXIPL (https://gtexportal.org/home/gene/MLXIPL (accessed on 12 May 2023)) are derived from GTEx Portal and presented with metabolic organs.

**Figure 2 ijms-24-08811-f002:**
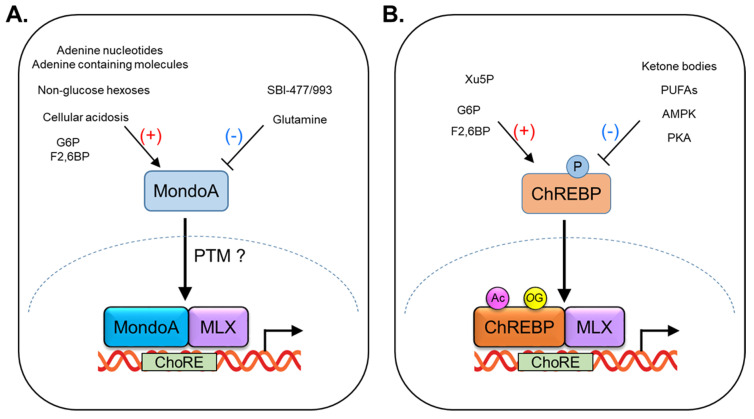
Regulation of MondoA and ChREBP activity by nutrients and post−translational modification (PTM). (**A**) MondoA transcriptional activity is elevated by glucose−6−phosphate (G6P), fructose-2,6−bisphosphate, non-glucose hexoses (allose, 3−O−methylglucose, and glucosamine), cellular acidosis, adenine nucleotides, and adenine−containing molecules, and is inhibited by glutamine and SBI−477/993. (**B**) ChREBP transcriptional activity is increased by G6P, F2,6BP, and xylulose−5−phosphate (Xu5P), and is decreased by ketone bodies, polyunsaturated fatty acids (PUFAs). Nuclear translocation of ChREBP is inhibited by its phosphorylation, regulated by PKA or AMPK. Acetylation (Ac) and O−GluNAcylation (OC) activate ChREBP transcriptional activity.

## Data Availability

All data used and analyzed in this study are included in this published article. Additional information will be available from the corresponding author upon reasonable request.

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
