# Peer review of "The Function of MondoA and ChREBP Nutrient—Sensing Factors in Metabolic Disease"

_ijms, 2023, doi:10.3390/ijms24108811_

Round 1

Reviewer 1 Report

Overall the manuscript is well design and have scientific merit. However, following points should be address

1.       English language and grammar need to be improved

2.       Please insert reference in Each row of Table-1

3.       How about the activities of MondoA and ChREBP in Kidney as SGLT2 play a pivotal role?

4.       Conclusion should be concise.

5.       Please provide take home message for the reader as well as future direction for the scientist.

Author Response

Response to Reviewer #1

I would like to take this opportunity to express my sincere gratitude for all the time and effort that you have invested in the review and evaluation of my manuscript. I also appreciate the editor and reviewers for their positive and constructive feedback on our paper entitled “The Function of MondoA and ChREBP Nutrient-Sensing Factors in Metabolic Disease”. Your careful comments and constructive feedback were invaluable as I revised the manuscript. Please see below, I have provided a point-by-point response to each reviewer's comments. Hopefully, the revised manuscript meets the reviewers' criteria, and I have made every effort to incorporate all suggestions into the manuscript.

  1. English language and grammar need to be improved

; The manuscript is updated per reviewer's comment. This manuscript has been edited and proofread by professionals belonging to Editage (editing service).

  1. Please insert reference in Each row of Table-1.

; Table 1 is updated with the references. Please check it.

  1. How about the activities of MondoA and ChREBP in Kidney as SGLT2 play a pivotal role?

; MondoA(Mlxip) and ChREBP(Mlxipl) are moderately expressed in Kidney compared to other tissues. The expression level of Txnip, which is known as an insulin signaling inhibitor, is quite high in Kidney. In addition, TXNIP plays a critical role in the regulation of NLRP3 inflammasome pathway which is also important in Kidney diseases. Interestingly, Txnip expression is directly regulated by MondoA and ChREBP. Unfortunately, so far there are no reports on the function of MondoA and ChREBP in the Kidney. I believe that MondoA and ChREBP could have specific roles in the Kidney. 

  1. Conclusion should be concise.

; The manuscript is updated per reviewer's comment.

  1. Please provide take home message for the reader as well as future direction for the scientist.

; The manuscript is updated per reviewer's comment.

Reviewer 2 Report

please see the attached file

Author Response

Response to Reviewer #2

I would like to take this opportunity to express my sincere gratitude for all the time and effort that you have invested in the review and evaluation of my manuscript. I also appreciate the editor and reviewers for their positive and constructive feedback on our paper entitled “The Function of MondoA and ChREBP Nutrient-Sensing Factors in Metabolic Disease”. Thank you again for your kind comments.
